# PHYSICALLY-GUIDED OPTICAL INVERSION ENABLE NON-CONTACT SIDE-CHANNEL ATTACK ON ISOLATED SCREENS

**Zhiwen Zheng**[1,2,3*]    **Yuheng Qiao**[1,2*]    **Xiaoshuai Zhang**[4*]    **Zhao Huang**[5]    **Tao Zhang**[1]
**Huiyu Zhou**[6]    **Shaowei Jiang**[1†]    **Jin Liu**[1,2,3†]    **Wenwen Tang**[7†]    **Xingru Huang**[1,2,3†]

[1] Hangzhou Dianzi University, Hangzhou 310018, China
[2] Weidian Corporation, Beijing 100015, China
[3] Zhejiang Provincial Key Laboratory of Low Altitude Ubiquitous Networking Technology, Hangzhou Dianzi University, Hangzhou, 310018, China
[4] Faculty of Information Science and Engineering, Ocean University of China, Qingdao 266404, China
[5] School of Natural and Computing Sciences, University of Aberdeen, Aberdeen AB24 3UE, Great Britain
[6] School of Computing and Mathematical Sciences, University of Leicester, Leicester LE1 7RH, UK
[7] Johns Hopkins University, Baltimore, MD 21218, USA

## ABSTRACT

Noncontact exfiltration of electronic screen content poses a security challenge, with side-channel incursions as the principal vector. We introduce an optical projection side-channel paradigm that confronts two core instabilities: (i) the near-singular Jacobian spectrum of projection mapping breaches Hadamard stability, rendering inversion hypersensitive to perturbations; (ii) irreversible compression in light transport obliterates global semantic cues, magnifying reconstruction ambiguity. Exploiting passive speckle patterns formed by diffuse reflection, our Irradiance Robust Radiometric Inversion Network (IR$^4$Net) fuses a Physically Regularized Irradiance Approximation (PRIrr-Approximation), which embeds the radiative transfer equation in a learnable optimizer, with a contour-to-detail cross-scale reconstruction mechanism that arrests noise propagation. Moreover, an Irreversibility Constrained Semantic Reprojection (ICSR) module reinstates lost global structure through context-driven semantic mapping. Evaluated across four scene categories, IR$^4$Net achieves fidelity beyond competing neural approaches while retaining resilience to illumination perturbations.

## 1 INTRODUCTION

Non-contact exfiltration of electronic screen information under unauthorized conditions represents a formidable challenge in information security. Long regarded as the ultimate safeguard, physical isolation may yet succumb to the merest reflection wall-scattered luminescence alone can betray sensitive content. This paper proposes a novel optical projection side-channel attack paradigm. Leveraging intrinsic optical characteristics, self-emissive targets enable imaging solely via their environmental projections. The resulting surveillance modality is passive and non-contact, with limited susceptibility to interception.An attacker can remotely capture the scattered light patterns projected onto nearby surfaces (e.g., walls) and use them to reconstruct the original screen content. As illustrated in Fig. 1, the attacker and the target remain physically separated, with no direct line-of-sight, no RF monitoring, and no communication link needed. This approach is highly stealthy and non-invasive, and it exposes new avenues of information leakage even in systems previously considered secure, such as laser protective glazing, electromagnetically shielded, or physically isolated environments.

Compare to traditional side-channel attacks, this optical approach leverages environmental media as a covert communication path. Microwave/electromagnetic techniques for tracking are vulnerable to attenuation and shielding; network-based channels are constrained by connectivity and congestion; hardware requirements are substantial; and active probing is readily detected, thereby revealing the

---

*Equal Contribution.
†Corresponding Authors.

operator's location.Electromagnetic-based attacks, for instance, rely on stray field emissions and are limited by distance, shielding, and ambient noise; Network-based attacks require connectivity and software vulnerabilities, making them inapplicable to air-gapped systems and often leaving traceable audit logs. In contrast, the optical projection side-channel attack proposed in this study circumvents these limitations, significantly enhancing attack feasibility and stealth, and prompting a fundamental reassessment of current defensive boundaries and strategies.

Despite its potential, this attack model presents substantial technical challenges. In everyday settings, self-luminous sources typically emit over a continuous spectrum, implying a continuously varying wavevector $k$. The corresponding Helmholtz solutions are therefore highly oscillatory, which makes it impossible to construct an accurate spatial propagation model. Furthermore, nonlinearity in the camera response undermines output stability and repeatability. The mapping from screen content to scattered speckles is ill-conditioned; the Jacobian matrix of the transformation has singular values that collapse in multiple directions, violating Hadamard's stability criterion. As a result, even minor irradiance fluctuations can be magnified into major structural distortions in the reconstructed image such as unpredictable edge displacement, false textures, or semantic drift. In addition, the inherently irreversible compression, along with occlusion, diffraction, and other optical effects, causes significant loss of global semantic structure and contextual cues. Without strong regularization, these losses manifest as blurry edges, disordered textures, and semantic discontinuities, leading to highly uncertain reconstructions.

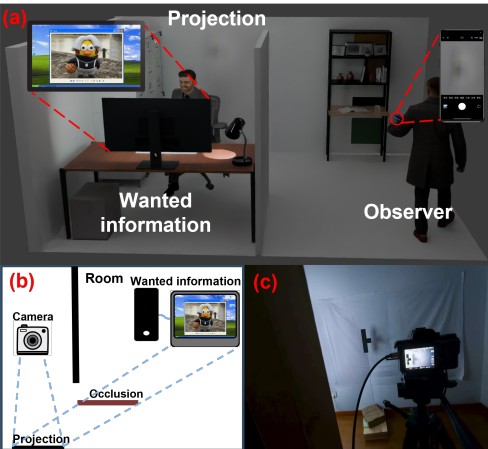

Figure 1: In the figure, (a), (b), and (c) correspond to the rendered scene, schematic diagram, and real-world scene respectively.

To overcome these challenges, we propose IR$^4$Net, a radiometric-inversion neural architecture that integrates physical modeling with deep learning priors, substantially improving the fidelity and stability of screen image reconstruction in optical side-channel scenarios. IR$^4$Net comprises Physically-Regularized Irradiance Approximation (PRIrr-Approximation) and Irreversibility-Constrained Semantic Re-Projection (ICSR). PRIrr-Approximation recasts nonlinear optical-field inversion as a learnable iterative path, embedding forward/reverse propagation physics through neural modules to yield an estimate consistent with irradiance constraints; by constraining the solution's trajectory, amplification of minute perturbations is curtailed. Residual noise sensitivity and detail loss from multi-scale diffraction persist, so a frequency-selective upsampling network decouples perturbations via a multi-scale frequency separation module, enabling hierarchical reconstruction from low-frequency contours to high-frequency details while damping inconsistent components. Finally, to mitigate irreversible semantic loss, ICSR builds a stable mapping in deep semantic space that aligns global structure with visual context, re-embedding abstract features under perceptual consistency constraints to infer and complete missing information.

The contribution of this paper are summarized as follow:

- To the best of our knowledge, this work is the first to demonstrate that diffuse wall reflections can serve as a viable optical side channel for reconstructing on-screen content,

and to propose the optical projection attack paradigm. This reveals a novel and previously overlooked avenue of information leakage in physically isolated environments.

- We introduce the PRIrr-Approximation module, reformulating optical field inversion as a physics-guided, learnable iterative trajectory to yield a stable initial estimate. A frequency-selective upsampling mechanism then drives progressive reconstruction from low to high frequencies, mitigating perturbation amplification and preserving structural integrity.

- We propose the ICSR module, which constructs a global-structure-aware semantic response within a deep semantic space. By embedding semantic features into a perceptual-consistency-constrained domain and applying context-driven completion rules to occluded and diffraction-corrupted regions, ICSR enhances edge continuity and semantic fidelity.

## 2 RELATED WORK

**Side-Channel Attacks(SCAs)** exploit electromagnetic, optical, acoustic, and microarchitectural leakages to infer display states. EM-based visual eavesdropping reconstructs HDMI video or camera views from unintended emanations and profiled traces Fernández et al. (2024); Long et al. (2024); Fang et al. (2022). Optical side channels turn commodity and ambient light sensors into imaging probes that recover scene or screen patterns from global illumination variations Chakraborty et al. (2017); Liu et al. (2024a). Acoustic and ultrasonic reflections around devices and robots encode passwords, keystrokes, and UI states under non-line-of-sight conditions Wang et al. (2024); Duan et al. (2024); Chen et al. (2024). Cache-based SCAs on DNN executables enable stealthy inference about processed visual content and internal architectures Liu et al. (2024b); Wang et al. (2022a); Gupta et al. (2023); Zhu et al. (2024), while broader models systematize cloud, biometric, and post-quantum leakage channels Albalawi et al. (2022); Johnson & Ward (2022); Ji & Dubrova (2023); Devi & Majumder (2021). However, existing optical SCAs typically rely on sensors co-located with the device or in direct view of the display, and none exploit diffuse wall reflections as an independent, remote optical side channel for recovering isolated screen content.

**Coherent Image generation** from structured priors integrates realism with domain constraints. Super-resolution He et al. (2022); Hong & Lee (2024); Chen et al. (2025), denoising Ye et al. (2025); Yang et al. (2025), and dehazing Ma et al. (2025); Fu et al. (2025); Wang et al. (2025) models reflect continuously improving efficacy Ryou et al. (2024). Transformer encoders such as Styleformer modulate diversity via attention-weighted embeddings Park & Kim (2022), while latent diffusion with implicit decoders enables scale-agnostic synthesis through multiresolution cascades Kim & Kim (2024). Patch tokenization fused with global context further boosts dehazing performance Ji-uchen Chen & Li (2025), and inter-channel consistency drives unsupervised deraining Dong et al. (2025). Recently, physics-guided approaches have incorporated forward models into dehazing, microscopy reconstruction, restoration of scattering-degraded images, and inverse rendering Lihe et al. (2024); Li et al. (2024); Qiao et al. (2025); Wu et al. (2025). However, the underlying physical assumptions in these models are tailored to specific transport or imaging/rendering mechanisms and are not well suited to capturing multi-scale diffraction and wavefront interference, making it difficult to recover occluded emissive patterns from strongly diffusive projections.

## 3 METHOD

Radiometric inversion under optical projection constitutes a severely ill-conditioned problem wherein nonlinear image-formation dynamics, perturbation amplification, and irreversible semantic degradation impede stable recovery. To address these challenges, we introduce the $IR^4Net$ (Fig 2) to integrate physical priors with learned optimization. First, PRIrr-Approximation formulates inversion as a physics-guided iterative trajectory embedding optical propagation operators with momentum-based updates to maintain consistency and mitigate cumulative error. A dual-path perturbation dissipation module concurrently performs spatial diffusion and semantic attenuation, while a frequency-selective multi-scale upsampling scheme constrains cross-scale energy propagation to reduce high-frequency amplification. Subsequently,ICSR establishes semantic completion and structural alignment within a perceptual space, enabling coherent reconstruction characterized by structurally preserved contours and contextually consistent textures.

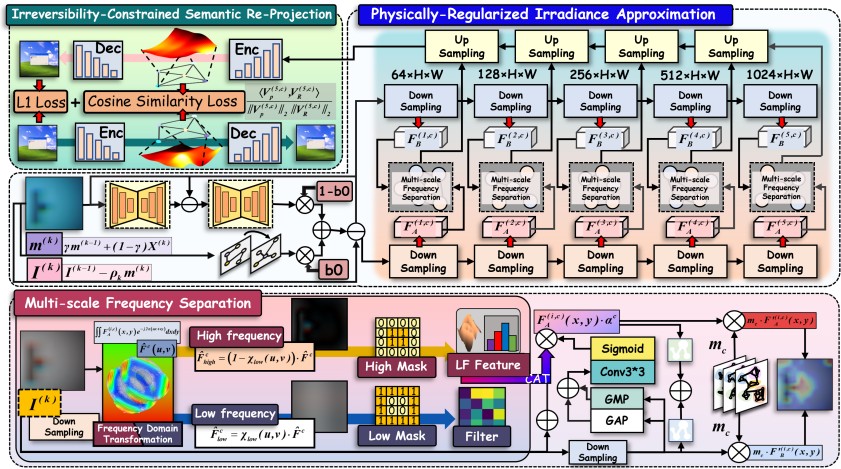

Figure 2: Overall architecture of IR$^4$Net, comprising the PRIrr-Approximation and ICSR modules. The multi-scale frequency separation module, a key component of ICSR, is implemented via concatenation.

## 3.1 Physically-Regularized Irradiance Approximation

Optical-projection side-channel attacks confront a fundamental challenge in the intricate physics of image formation: the observed image arises from a highly nonlinear mapping of the original irradiance through successive diffraction, scattering and reflection. This process imposes extreme information compression and yields an operator whose singular values tend toward zero, so that infinitesimal irradiance perturbations at the input become dramatically amplified in inversion, inducing severe distortion and instability. To mitigate this, we introduce a physics-constrained module: guided inversion trajectory embeds optical-propagation modeling to guarantee physical consistency; in parallel, a frequency-selective upsampling network decouples perturbations and reconstructs multi-scale spectral components, structurally suppressing their amplification.

Our scheme models optical effects via a transfer operator $\Phi(\cdot)$, and derives its inverse approximation $\Psi(\cdot)$ to harvest feedback. A momentum initialization melds local priors with multi-scale global feedback, steering each iteration along coherent directions. Momentum-guided gradient updates suppress noise and error accumulation, yielding feature estimates $\hat{I}^{(k)}$ that converge toward an inversion of the source radiance; derivations reside in appendix.

In the dual-path feature-dissipation stage, we deploy a frequency-selective upsampling network in parallel with spatial diffusion and semantic attenuation pathways to capture the rapid amplification of minute screen-to-wall perturbations. This decoupled architecture structurally restrains perturbation growth and disperses its energy, to maintain robustness against projection-induced distortions.

The input $I^{(k)}$ flows through two paths: the spatial diffusion path applies a second-order differential kernel to the local gradient:

$$F_A^{(i,c)}(x,y) = \phi\Big(\iint_{B_r} \kappa_A^{(i,c)}(\xi,\eta)\, \frac{\partial^2 I^{(k)}}{\partial x \partial y}(x-\xi, y-\eta)\, \mathrm{d}\xi\, \mathrm{d}\eta + b_A^{(i,c)}\Big). \quad (1)$$

Here, $I^{(k)}$ is the feature map at iteration $k$; $i$ and $c$ denote decoder and channel indices; $\frac{\partial^2 I^{(k)}}{\partial x \partial y}$ is the mixed second-order derivative, capturing local curvature; $B_r$ the neighborhood centered at $(x,y)$ with radius $r$; $\kappa_A^{(i,c)}(\xi,\eta)$ the second-order differential kernel for decoder $i$, channel $c$; $b_A^{(i,c)}$ the bias term; $\phi(\cdot)$ the activation; and $F_A^{(i,c)}(x,y)$ the spatial diffusion output.

The semantic attenuation path, through an attention mechanism, mitigates disturbance components in the semantic dimension, where for the $i$-th attention head, the linear projection is given by

$$(Q^{(i)}, K^{(i)}, V^{(i)})(x) = I^{(k)}(x)\,(W_Q^{(i)}, W_K^{(i)}, W_V^{(i)}) \quad (2)$$

$$A^{(i)}(x, x') = \frac{\exp\langle Q^{(i)}(x),\, K^{(i)}(x')\rangle}{\displaystyle\int_{\Omega_s} \exp\langle Q^{(i)}(x),\, K^{(i)}(x')\rangle\, dx'} \tag{3}$$

$$F_B^{(i,c)}(x, y) = \phi\left( \int_{\Omega_s} A^{(i)}(x, x')\, V^{(i,c)}(x')\, dx' + b_B^{(i,c)} \right) \tag{4}$$

In this context, $W_Q^{(i)}, W_K^{(i)}, W_V^{(i)} \in \mathbb{R}^{C \times d}$ represent the projection matrices for query, key, and value, with $C$ being the original number of feature channels and $d$ the projected dimension. The attention mechanism disperses disturbance components in the semantic space to ensure that the disturbance does not concentrate spatially. Consequently, $F_B^{(i,c)}(x, y)$ represents the output feature of this semantic attenuation path.

Subsequently, the multi-scale frequency separation module concatenates the outputs of both paths in the spatial domain and performs gating in the frequency domain. This step guarantees that only low-frequency components with cross-scale consistency and structural robustness are amplified layer by layer, while high-frequency components that lack scale consistency attenuate during propagation.

$$\widehat{F}^c(u, v) = \iint F_A^{(i,c)}(x, y)\, e^{-j2\pi(ux+vy)}\, dx\, dy, \tag{5}$$

where, $\widehat{F}^c(u, v)$ represents the frequency-domain transformation of the concatenated features, and $u$ and $v$ are the spatial frequency coordinates along the $x$- and $y$-axes.

$$(F_{\text{low}}^c, F_{\text{high}}^c) = \left( \mathcal{F}^{-1}[\chi_{\text{low}}\widehat{F}^c],\ \mathcal{F}^{-1}[(1 - \chi_{\text{low}})\widehat{F}^c] \right). \tag{6}$$

The channel response $\alpha^c$ passes through adaptive gating to fuse low/high-frequency features, enabling cross-scale propagation; final fusion is:

$$\alpha^c = \sigma\left( W_2\, \phi\big(W_1 \int_{\Omega_s} \big|\nabla F_{\text{low}}^c + \nabla F_{\text{high}}^c\big|\, dx\, dy\big) \right) \tag{7}$$

$$F_A'^{(i,c)}(x, y) = F_A^{(i,c)}(x, y)\left(1 + \alpha^c\right) \tag{8}$$

This attention-based fusion ensures a balanced contribution from both spatial diffusion and semantic attenuation, with each component adapting based on the gradient magnitude of low- and high-frequency features. Thus, $F_A'^{(i,c)}(x, y)$ represents the feature after weighted fusion.

The channel attention weight $m_c$ is generated through global average pooling and differential operations, with the calculation formula given as:

$$m_c = \sigma\left( \frac{d}{d\hat{g}_c} \left( \int_{x=0}^{H} \int_{y=0}^{W} (\hat{g}_c \cdot g_c(x, y))\, dx\, dy \right) \right). \tag{9}$$

Here, $g_c(x, y)$ is the value at position $(x, y)$ of the $c$-th channel in the input feature map, and $\hat{g}_c$ is its global average. $m_c$ denotes the attention weight for channel $c$, and $\sigma$ is the Sigmoid function ensuring $m_c \in [0, 1]$. Using $m_c$, the feature maps $F_A'^{(i,c)}(x, y)$ and $F_B^{(i,c)}(x, y)$ are fused at each $(x, y)$ to produce the output map $\widetilde{F}^c(x, y)$:

$$\widetilde{F}^c(x, y) = m_c \cdot \left( F_A'^{(i,c)}(x, y) + F_B^{(i,c)}(x, y) \right). \tag{10}$$

In this equation, $F_A'^{(i,c)}(x, y)$ and $F_B^{(i,c)}(x, y)$ represent the values of the $c$-th channel of the input feature maps $F_A'$ and $F_B$ at position $(x, y)$. Through this weighted fusion process, the final output feature map $\widetilde{F}^c(x, y)$ incorporates the fused channel information.

Following this, multi-head attention mechanisms are employed to capture and suppress any remaining perturbation structures within the fused features $\widetilde{F}^c$:

$$A_h(x, x') = \exp \langle \partial_x Q_h(x),\, K_h(x') \rangle + \langle Q_h(x),\, \partial_{x'} K_h(x') \rangle \,, \tag{11}$$

$$O_h(x) = \int_{\Omega_s} A_h(x, x')\, V_h(x')\, dx' \,, \tag{12}$$

$$Z^{(i)} = \text{Concat}_h \left( O_h(x) \right) + \widetilde{F}^{(c)}(x, y) \,. \tag{13}$$

In this context, the space-derivative mappings of each attention head allow for precise quantification of perturbation effects on attention distribution. Residual connections preserve stable structural information throughout the process.

During the multi-scale frequency-selective upsampling and output synthesis stage, perturbation growth along successive upsampling layers is mitigated through hierarchical decomposition and reconstruction of enhanced features $Z^{(i)}$ with the preceding layer $Z^{(i-1)}$. Specifically, the intermediate interpolation $U^{(i)}(x)$ is computed as:

$$U^{(i)}(x) = \iint Z^{(i)}(x') \prod_{j=1}^{2} \max\big(0, 1 - |x_j - x'_j|\big)\, dx', \tag{14}$$

and the upsampled representation $F_{\text{up}}^{(i)}(x)$ is expressed as:

$$F_{\text{up}}^{(i)}(x) = \phi\left( \iint \kappa_{\text{up}}^{(i)}(x, x') \big[ U^{(i)}(x') + Z^{(i-1)}(x') \big]\, dx' \right), \tag{15}$$

where the bilinear interpolation kernel $\prod_{j=1}^{2} \max(0, 1 - |x_j - x'_j|)$ operates in concert with the learned upsampling kernel $\kappa_{\text{up}}^{(i)}$, enabling progressive reconstruction. This hierarchical scheme introduces information from low to high frequencies in a controlled manner to permit expansion only of cross-scale-consistent structural features when attenuating perturbations lack multi-scale support.

The final stage maps the first-level upsampled feature into the pixel domain via an output convolution with kernel $\kappa_{\text{out}}$ and bias $b_{\text{out}}$:

$$\widehat{\mathbf{J}}^{(k)}(x, y) = \iint \kappa_{\text{out}}(x, x') F_{\text{up}}^{(1)}(x')\, dx' + b_{\text{out}}. \tag{16}$$

Through the integration of physical constraints with frequency-selective hierarchical fusion, this mechanism is designed to suppress propagation of fine-scale irradiance perturbations and maintain structural consistency during reconstruction of the projected image.

## 3.2 IRREVERSIBILITY-CONSTRAINED SEMANTIC RE-PROJECTION

The inversion of optical projection requires irreversible, high-compression mapping original imagery. However, it suffers severe loss of global semantic structure and visual context, manifesting as edge blur, texture artifacts and semantic misalignment due to occlusion, diffraction and reflection. To address this challenge, we introduce the ICSR, comprising two parallel modules: a primary mapping network, driven by a prior-guided map, devoted to restoration of low-level structural detail; and a collaborative completion network, which extracts stable abstract semantic embeddings from the projected observation to capture global semantics and contextual cues. Building upon these, a stable mapping from semantic to structural space is established to enable high-dimensional semantic features to be dynamically fed back into the primary network's representation domain, thereby enforcing constrained completion and inference over missing regions. Here, the primary network's structural-space mapping features are $V_P^{(5,c)}(x, y) \in \mathbb{R}^d$ and the abstract semantic-space features are $V_R^{(5,c)}(x, y) \in \mathbb{R}^d$ where $c$ denotes input channels, $5$ denotes the encoder stage, $(x, y)$ spatial coordinates and $d$ the feature dimension; derivation appears in appendix.

$$\mathbf{v}_P^{(5,c)}(x, y) = \left( v_{P,1}^{(5,c)}(x, y),\, v_{P,2}^{(5,c)}(x, y),\, \ldots,\, v_{P,d}^{(5,c)}(x, y) \right), \tag{17}$$

$$\mathbf{v}_R^{(5,c)}(x,y) = \left( v_{R,1}^{(5,c)}(x,y), v_{R,2}^{(5,c)}(x,y), \ldots, v_{R,d}^{(5,c)}(x,y) \right). \tag{18}$$

In order to preserve the consistency between the semantic and structural feature spaces, to prevent semantic drift, and to enhance the stability of the completion inference process, we compute the cosine similarity between the projected features as follows:

$$\mathrm{CosSim}(x,y) = \frac{\sum_{i=1}^{d} v_{P,i}^{(5,c)}(x,y)\, v_{R,i}^{(5,c)}(x,y)}{\sqrt{\sum_{i=1}^{d} \left( v_{P,i}^{(5,c)}(x,y) \right)^2 + \epsilon}\, \sqrt{\sum_{i=1}^{d} \left( v_{R,i}^{(5,c)}(x,y) \right)^2 + \epsilon}} \tag{19}$$

where $\epsilon > 0$ is introduced to prevent division by zero.

Subsequently, for each batch $\mathcal{B} = \{(x_j, y_j)\}_{j=1}^{N}$, the batch loss function is defined as:

$$s_j = \frac{\sum_{i=1}^{d} v_{P,i}^{(5,c)}(x_j,y_j)\, v_{R,i}^{(5,c)}(x_j,y_j)}{\sqrt{\sum_{i=1}^{d} (v_{P,i}^{(5,c)}(x_j,y_j))^2 + \epsilon}\, \sqrt{\sum_{i=1}^{d} (v_{R,i}^{(5,c)}(x_j,y_j))^2 + \epsilon}}. \tag{20}$$

$$\mathcal{L}_{\mathrm{batch}} = \frac{1}{N} \sum_{j=1}^{N} (1 - s_j)^{\alpha} + \lambda \|\Theta\|_2^2. \tag{21}$$

where $\lambda \| \Theta \|_2^2$ represents the L2 regularization term.

This loss leverages multi-scale semantic alignment to improve missing-region completion, yielding sharp, realistic, and coherent reconstructions.

## 4 EXPERIMENTS

Four datasets: ReSh-WebSight, ReSh-Password, ReSh-Chart, and ReSh-Screen were employed to emulate user-interface layouts, password-entry interfaces, chart renderings, and desktop scenarios, randomized into training, validation, and test subsets in an 8:1:1 ratio. All experiments were implemented in PyTorch on an NVIDIA RTX 3090 GPU cluster, using Adam optimizer with a fixed learning rate of $1 \times 10^{-4}$ and a batch size of 16; other hyperparameters were set to their default values unless stated otherwise.

| Methods | Source | ReSh-WebSight | | | ReSh-Password | | | ReSh-Screen | | | ReSh-Chart | | |
|---|---|---|---|---|---|---|---|---|---|---|---|---|---|
| | | PSNR↑ | RMSE↓ | SSIM↑ | PSNR↑ | RMSE↓ | SSIM↑ | PSNR↑ | RMSE↓ | SSIM↑ | PSNR↑ | RMSE↓ | SSIM↑ |
| HVI-CIDNet | CVPR,25 | 18.940 | 33.837 | 0.792 | 13.024 | 57.269 | 0.858 | 21.027 | 26.686 | 0.708 | 15.720 | 44.843 | 0.692 |
| DarkIR | CVPR,25 | 19.234 | 32.587 | 0.779 | 13.580 | 53.883 | 0.855 | 21.609 | 25.215 | 0.706 | 16.861 | 39.011 | 0.709 |
| AST | CVPR,24 | 19.502 | 31.026 | 0.787 | 14.022 | 51.199 | 0.832 | 21.574 | 24.823 | 0.673 | 16.909 | 38.515 | 0.709 |
| ConvIR | CVPR,24 | 19.573 | 30.678 | 0.799 | 14.779 | 47.077 | 0.867 | 22.010 | 23.718 | 0.731 | 16.678 | 39.569 | 0.707 |
| C2PNet | CVPR,23 | 15.641 | 52.514 | 0.769 | 11.209 | 70.458 | 0.813 | 16.428 | 44.883 | 0.552 | 15.278 | 46.885 | 0.666 |
| Uformer | CVPR,22 | 19.698 | 30.262 | 0.798 | 14.142 | 50.578 | 0.874 | 22.299 | 22.885 | 0.725 | 17.068 | 39.087 | 0.712 |
| UNet | MICCAI,15 | 17.735 | 38.744 | 0.764 | 11.891 | 65.055 | 0.827 | 20.195 | 28.114 | 0.664 | 16.133 | 42.120 | 0.682 |
| BicycleGAN | NIPS,17 | 18.680 | 35.453 | 0.775 | 9.784 | 82.939 | 0.781 | 18.305 | 46.888 | 0.600 | 15.289 | 48.376 | 0.632 |
| DivCo | CVPR,21 | 13.353 | 66.098 | 0.721 | 9.266 | 87.803 | 0.730 | 12.280 | 72.146 | 0.369 | 11.091 | 76.426 | 0.523 |
| pix2pix | CVPR,17 | 13.582 | 62.361 | 0.651 | 8.146 | 99.907 | 0.684 | 8.043 | 103.377 | 0.232 | 12.475 | 63.885 | 0.452 |
| CycleGAN | ICCV,17 | 13.068 | 66.529 | 0.680 | 6.206 | 124.912 | 0.601 | 10.358 | 89.134 | 0.316 | 12.348 | 67.200 | 0.494 |
| IR⁴Net | Ours | **20.708** | **26.719** | **0.820** | **15.030** | **45.911** | **0.887** | **25.812** | **16.531** | **0.817** | **17.363** | **36.748** | **0.731** |

Table 1: Quantitative comparison of IR⁴Net against reconstruction-centric methods (HVI-CIDNetYan et al. (2025),DarkIRFeijoo et al. (2025),ASTZhou et al. (2024),ConVIRCui et al. (2024),C2PNetZheng et al. (2023),UformerWang et al. (2022b), UNetRonneberger et al. (2015)) and generation-centric methods (BicycleGANZhu et al. (2017b),DivcoLiu et al. (2021),pix2pixIsola et al. (2017),CycleGANZhu et al. (2017a)) on four benchmarks (ReSh-WebSight, ReSh-Password, ReSh-Screen, ReSh-Chart) under identical data splits and optimization.

## 4.1 COMPARATIVE EVALUATION

Deployed across four canonical benchmarks ReSh-WebSight, ReSh-Password, ReSh-Screen, and ReSh-Chart for assessment under disparate projection scenarios, IR$^4$Net is juxtaposed with reconstruction-centric (Uformer, ConvIR, UNet) and generation-centric (pix2pix, CycleGAN, BicycleGAN) counterparts, each trained and tested under identical data partitions and optimisation regimes. Table 1 reports results on PSNR, RMSE, and SSIM: Specifically, PSNR on ReSh-Screen increases by 15.7% relative to Uformer, and RMSE on ReSh-WebSight falls by 13.9% compared to AST. Qualitative illustrations (Fig 3) reveal more consistent restoration of edges, textures, and occluded regions. Such behaviour likely reflects the interplay of two structural modules: PRIrr-Approximation, embedding physics-consistent, momentum-guided inversion trajectories with frequency-selective perturbation dissipation, and ICSR, enacting a stable semantic-space mapping to align and replenish irreversibly lost information. In both perceptual and physical domains, these mechanisms operate in concert to sustain reconstruction fidelity and robustness.

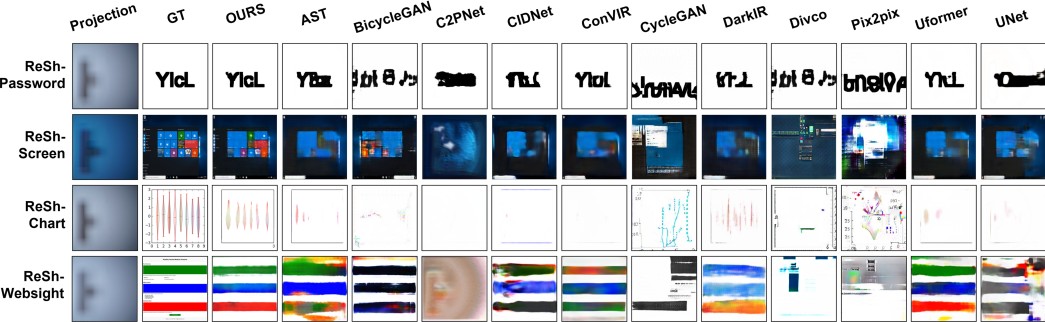

Figure 3: Visual comparison of IR$^4$Net and baseline methods on four datasets. Our model yields visually more faithful restorations across various scenes.

## 4.2 ABLATION EXPERIMENT

Inversion behaviour was evaluated across three datasets using four iterative schemes: classical momentum formulations including ADMM, NAG, and Heavy-Ball, and the proposed update strategy. Table 2 reports the performance under PSNR, SSIM, RMSE,and LPIPS;This variation may derive from a dual coupling design: structure-aware momentum initialization, achieved through a learnable convolutional operator over local receptive fields, yielding priors aligned with intrinsic structural patterns; and a physics-feedback pathway, where inverse approximations are constructed from encoded residuals to capture projection-induced perturbations to constrain the update direction in physically admissible regimes. Residual-gated dynamic weighting integrates these cues to mitigate error amplification introduced by near-singular transmission operators while accumulated momentum smooths the update trajectory. Stability observed under diverse conditions suggests adaptability in high-compression, nonlinear inversion scenarios. Additional ablation studies are provided in appendix.

| Metric | Chart | | | | Screen | | | | WebSight | | | |
|---|---|---|---|---|---|---|---|---|---|---|---|---|
| | OURS | ADMM | NAG | Heavyball | OURS | ADMM | NAG | Heavyball | OURS | ADMM | NAG | Heavyball |
| PSNR↑ | **17.363** | 17.180 | 17.214 | 17.192 | **25.812** | 25.155 | 25.090 | 25.077 | **20.708** | 20.707 | 20.621 | 20.533 |
| RMSE↓ | **36.748** | 37.367 | 37.308 | 37.447 | **16.531** | 17.680 | 17.672 | 17.754 | **26.719** | 27.024 | 27.349 | 27.629 |
| SSIM↑ | **0.731** | 0.725 | 0.724 | 0.724 | **0.817** | 0.806 | 0.808 | 0.808 | **0.820** | 0.808 | 0.808 | 0.807 |
| LPIPS↓ | **0.431** | 0.468 | 0.465 | 0.462 | **0.216** | 0.232 | 0.235 | 0.231 | **0.282** | 0.299 | 0.299 | 0.300 |

Table 2: Performance comparison of four iterative schemes across three datasets.

### 4.3 Luminance Robustness Evaluation

To assess stability under low illumination, an experimental setup was devised where display luminance was progressively attenuated to emulate irradiance decay in real projection scenarios. Experiments were conducted on the four previously mentioned datasets, with screen brightness reduced by 0–300 nits. PSNR values were recorded for each method at incremental luminance levels.

As summarized in Table 3, pronounced performance degradation emerged for several baselines under reduced brightness. For instance, UNet exhibited a PSNR decline of approximately 68% on ReSh-Screen, whereas the proposed architecture registered a reduction of 25.9% under identical conditions. Visual evidence Fig 4 indicates that when luminance decreased, competing methods produced outputs with structural misalignment and blurred contours, while the proposed approach maintained coherent edge geometry and stable texture patterns. Results for other datasets, together with qualitative exemplars, appear in appendix.

| Model | 0 | 25 | 50 | 75 | 100 | 125 | 150 | 175 | 200 | 225 | 250 | 275 | 300 |
|---|---|---|---|---|---|---|---|---|---|---|---|---|---|
| **OURS** | **25.812** | **25.726** | **25.702** | **25.634** | **25.533** | **25.288** | **24.990** | **24.537** | **24.016** | **23.220** | **22.306** | **20.983** | **19.136** |
| **UNet** | 20.195 | 6.302 | 6.461 | 6.121 | 6.546 | 6.104 | 6.301 | 7.015 | 6.089 | 6.686 | 6.257 | 6.144 | 6.412 |
| **C2PNet** | 16.428 | 15.913 | 14.951 | 13.452 | 12.311 | 11.520 | 11.195 | 10.948 | 10.542 | 10.039 | 9.666 | 9.370 | 9.144 |
| **DarkIR** | 21.609 | 21.360 | 20.660 | 19.288 | 17.423 | 15.355 | 13.672 | 12.481 | 11.491 | 10.614 | 10.077 | 9.686 | 9.424 |
| **CIDNet** | 21.027 | 20.808 | 20.337 | 19.366 | 18.118 | 16.576 | 15.131 | 13.853 | 12.791 | 11.739 | 10.913 | 10.192 | 9.745 |
| **ConvIR** | 22.010 | 21.824 | 21.480 | 20.601 | 19.223 | 17.698 | 16.215 | 14.885 | 13.662 | 12.537 | 11.707 | 10.991 | 10.435 |

Table 3: PSNR comparison of different models under screen brightness reductions (in nits).

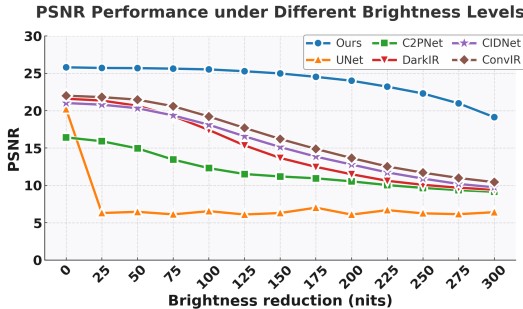

Figure 4: As screen brightness decreases on the ReSh-Screen dataset, our model's PSNR degrades significantly less than that of other methods.

These observations indicate that robustness to luminance attenuation arises from three architectural constraints: (i) a physics-regularized propagation path limiting perturbation diffusion; (ii) a frequency-selective hierarchical upsampling scheme ensuring cross-scale consistency; and (iii) a semantic-stability module restoring global context via feature-space completion. Without these constraints, conventional models suffer error amplification, causing structural collapse under low-intensity conditions. In contrast, the proposed design suppresses perturbations through physics-guided modeling, applies frequency-domain gating to limit non-structural energy propagation, and employs context-consistent semantic inference to recover projection-induced information loss. Together, these mechanisms preserve texture fidelity and ensure controlled, monotonic degradation across the luminance continuum.

### 4.4 Geometric Robustness under Camera Motion and Distance

Fig. 5 illustrates three planar-wall camera-motion settings. We evaluate orbital motion along horizontal and vertical arcs with the optical axis facing the wall, in-place rotation by varying pitch, yaw, and roll at a fixed camera center, and distance variation by translating the camera along the optical axis. Table 4 reports IR$^4$Net results for these perturbations in the left, middle, and right blocks.

Across all settings, IR$^4$Net remains stable. Under orbital motion, PSNR exceeds 21 dB for all poses and FID stays within 0.90 to 1.05. With increasing rotation, PSNR and SSIM decrease smoothly,

| Orbital motion | | | | | Camera rotation | | | | | Camera–wall distance | | | | |
|---|---|---|---|---|---|---|---|---|---|---|---|---|---|---|
| Pose (horizontal,vertical) | PSNR | SSIM | LPIPS | FID | Angle (pitch,yaw,roll) | PSNR | SSIM | LPIPS | FID | Dist. (m) | PSNR | SSIM | LPIPS | FID |
| (0,5) | 25.046 | 0.801 | 0.226 | 1.047 | (0, 0, 0) | 25.812 | 0.817 | 0.216 | 0.967 | 2 | 25.810 | 0.817 | 0.216 | 0.970 |
| (0,15) | 22.814 | 0.749 | 0.263 | 0.953 | (2, 0, 0) | 25.586 | 0.814 | 0.219 | 0.995 | 3 | 25.791 | 0.816 | 0.217 | 0.967 |
| (0,-5) | 25.267 | 0.807 | 0.224 | 0.997 | (5, 3, 2) | 24.508 | 0.793 | 0.234 | 1.057 | 4 | 25.764 | 0.816 | 0.217 | 0.975 |
| (10,0) | 22.901 | 0.740 | 0.263 | 0.899 | (8, 5, 4) | 21.470 | 0.719 | 0.285 | 0.996 | 5 | 25.690 | 0.815 | 0.218 | 0.980 |
| (15,0) | 21.275 | 0.705 | 0.294 | 0.986 | (0, -10, -3) | 19.906 | 0.697 | 0.308 | 1.006 | 6 | 25.541 | 0.813 | 0.219 | 0.990 |

Table 4: Summary of IR$^4$Net performance under three geometric perturbations: (1) orbital motion, (2) camera rotation, and (3) camera–wall distance variation. Each block lists five representative conditions from the full experiments.

yet at 8°, 5°, and 4° the method still reaches 21.47 dB PSNR and 0.719 SSIM without structural degradation. Varying the camera–wall distance from 2 m to 6 m causes only minor changes, with PSNR 25.81 to 25.54 dB and SSIM 0.817 to 0.813, indicating limited sensitivity to distance-related irradiance decay and speckle-scale variation.

This robustness stems from the combination of PRIrr-Approximation, which regularizes inversion using an irradiance-consistent propagation model and frequency-selective upsampling, and ICSR, which promotes semantic and structural consistency to mitigate information loss in projected speckle patterns. Additional results are provided in appendix.

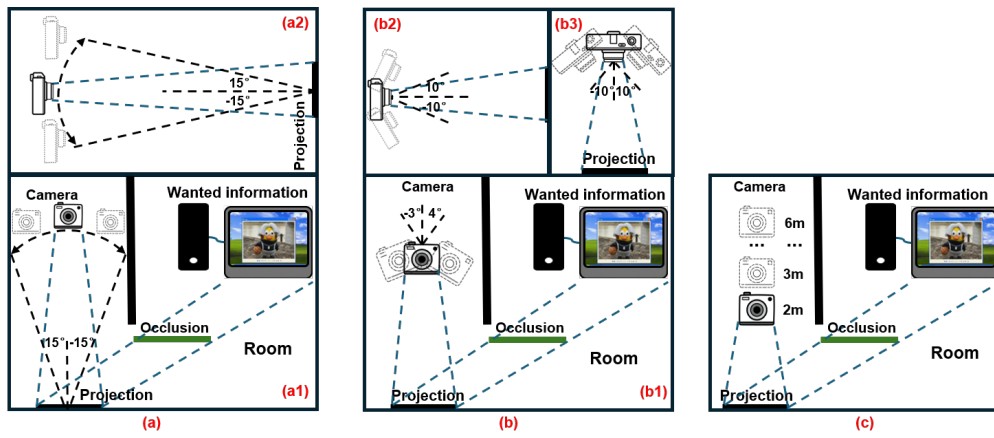

Figure 5: Experimental setups for camera motion. (a) Camera orbital motion: (a1) motion along a horizontal arc parallel to the ground; (a2) motion along a vertical arc perpendicular to the ground (side view). (b) Camera rotation: (b1) tilting parallel to the projection wall; (b2) tilting perpendicular to the ground (side view); (b3) tilting perpendicular to the projection plane (top view). (c) Camera–wall distance: camera translating along the direction normal to the projection plane.

## 5 CONCLUSION AND DISCUSSION

Non-contact exfiltration of screen content in physically isolated or shielded environments is achieved via an optical-projection side channel, realized by IR$^4$Net, a physics-constrained reconstruction framework embedding irradiance-consistent modeling and spectral regulation. Addressing two core challenges, namely nonlinear mapping ill-conditioning and semantic attrition, this architecture invalidates the notion that an air gap guarantees security. In the inversion-path stage, PRIrr-Approximation reformulates optical-field inversion as a learnable iterative trajectory that integrates forward and reverse propagation physics, mitigating perturbation amplification. In the spectral domain, a multi-scale frequency separation module decouples and hierarchically restores spectral components to reinforce cross-scale structural coherence and suppress noise. Furthermore, ICSR's abstract semantic-space mapping drives global semantic completion to bridge gaps induced by strong projection compressions. Experimental results demonstrate stable content restoration under attenuated irradiance, with SSIM and related metrics exceeding those of existing end-to-end models, to confirm the effectiveness and robustness of the physics-prior and deep-model fusion.

## 6 ACKNOWLEDGEMENT

This work was supported in part by the Young Scientists Fund of the National Natural Science Foundation of China under Grant 62506103 and 42501545; in part by the Fundamental Research Funds for the Provincial Universities of Zhejiang under Grant GK259909299001-026; in part by the Excellent Young Scientists Fund Program (Overseas) of Shandong Province under Grant 2025HWYQ-033; and in part by the China Postdoctoral Science Foundation (General Program) under Grant 2024M760716 and the Special Financial Grant from the China Postdoctoral Science Foundation under Grant 2025T180962.

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
