# OpenReview forum: "Physically-Guided Optical Inversion Enable Non-Contact Side-Channel Attack on Isolated Screens"
_ICLR.cc/2026/Conference — ICLR 2026 Poster_

### Official Review · Reviewer_4VYg · 2025-10-28

**Soundness:** 3
**Presentation:** 3
**Contribution:** 3
**Rating:** 6
**Confidence:** 3

**Summary:**

This paper introduces a novel optical projection side-channel attack that enables non-contact, passive reconstruction of screen content from diffuse wall reflections—posing a new threat to physically isolated systems. The core technical challenges are the ill-conditioned nature of the optical inversion (due to near-singular Jacobians and Hadamard instability) and irreversible semantic loss from light transport compression. To address these, the authors propose **IR4Net**, a physically guided deep inversion framework comprising two key modules: (1) **PRIrr-Approximation**, which embeds the radiative transfer equation into a learnable, momentum-guided iterative optimizer to stabilize inversion and suppress perturbation amplification; and (2) **ICSR (Irreversibility-Constrained Semantic Reprojection)**, which recovers global semantics via context-aware feature mapping in deep semantic space. Evaluated on four custom datasets (ReSh-WebSight, ReSh-Password, ReSh-Chart, ReSh-Screen), IR4Net outperforms both reconstruction-based (e.g., Uformer, ConvIR) and generative (e.g., CycleGAN) baselines in PSNR, SSIM, and RMSE, while demonstrating superior robustness under low illumination and noise.

**Strengths:**

(1) The paper identifies and demonstrates a previously overlooked attack vector—reconstructing screen content from *diffuse wall reflections alone*—which fundamentally challenges assumptions about the security of air-gapped systems.
(2) The proposed IR4Net integrates physics-based modeling (radiative transfer, Lambertian reflection) with deep learning in a principled way, using momentum-guided inversion and frequency-selective upsampling to address ill-posedness and perturbation sensitivity.
(3) Comprehensive experiments across four diverse datasets, ablation studies on optimization schemes, and robustness tests under low luminance and noise convincingly validate the method’s superiority over strong baselines.
(4) The ICSR module offers a creative solution to semantic loss by enforcing perceptual consistency between structural and semantic feature spaces, enabling coherent global reconstruction even when high-frequency details are irreversibly degraded.

**Weaknesses:**

- The datasets (e.g., ReSh-Screen with only 1,272 images) are not publicly released, and critical experimental details—such as exact camera-screen-wall geometry, wall BRDFs, or lighting calibration—are insufficiently described, severely limiting reproducibility.
- The method assumes ideal Lambertian wall reflectance, yet real-world surfaces often exhibit non-Lambertian or spatially varying reflectance; the paper lacks quantitative analysis of performance degradation under such conditions.
- The claim of “no line-of-sight” is potentially misleading: the setup still requires indirect optical paths (e.g., shared wall), and the attack’s feasibility under more extreme occlusion (e.g., around corners or through multiple bounces) remains unverified.

**Questions:**

(1) Could the authors provide the exact physical configuration used in data collection (e.g., screen-to-wall distance, camera-to-wall distance, wall material properties, ambient lighting conditions)? Without these, independent replication is nearly impossible.
(2) How does IR4Net perform on non-Lambertian surfaces (e.g., glossy paint, metal, or textured wallpaper)? Are there quantitative metrics (e.g., PSNR drop) showing sensitivity to BRDF deviation from the ideal diffuse assumption?
(3) The paper states the attack works “without line-of-sight,” but Figure 5(c) shows a shared wall between rooms. Can the method reconstruct content when the screen is not optically connected to the observed wall via any single-bounce path (e.g., around a corner with two or more diffuse bounces)?
(4) The ICSR module aligns structural and semantic features via cosine similarity—could this lead to plausible but factually incorrect hallucinations (e.g., generating “admin” instead of the actual typed password)? Are there examples of such failures?
- What are the computational requirements (inference time, memory footprint) of IR4Net? Is real-time reconstruction feasible on consumer hardware, or is the method limited to offline analysis?
- Why were recent physics-informed neural networks (e.g., PINNs, NeRF-based inverse rendering) not included in the baselines? How does IR4Net conceptually and empirically differ from these approaches?
(5) The ablation in Table 2 only compares iterative solvers but does not isolate the contributions of the spatial diffusion path, semantic attenuation path, and frequency-selective upsampling in PRIrr-Approximation. Could the authors provide a module-wise ablation?
(6) Given the serious security implications, have the authors engaged in responsible disclosure with relevant hardware or OS vendors? Do they plan to delay public release of code or datasets until practical countermeasures (e.g., screen filters, ambient light control) are developed?

---

> ### Author Response · Authors · 2025-11-24
> **Response to Reviewer 4VYg Regarding Weaknesses on Dataset Reproducibility, Non-Lambertian Robustness, Line-of-Sight Clarification, ICSR Semantic Reliability, Computational Requirements and Baseline Selection, and Module-wise Ablation with Responsible Disclosure**
>
> **(1) Exact Physical Configuration & Reproducibility**
>
> **Physical Configuration:** The experimental setup is **not** a "sweet-spot" geometry but a standard office simulation.
> * **Geometry:** Single diffuse-bounce (screen → wall → camera).
> * **Distances:** Screen-wall **0.9m**; Camera-wall **2.0m** (extended to **6m** in Table R4-3).
> * **Hardware:** Sony A7S II. **No** HDR or long-exposure tricks.
> * **Environment:** Indoor room, blinds closed, dim ambient light (desk lamp).
> * **Imaging configuration:**  Table 1-2 demonstrates that our system is tolerant to camera pose variations, and Appendix A.10 further confirms that even a partial observation of the wall speckle pattern is sufficient to recover useful on-screen information.
> * **Wall:** Matte white wall. Non-Lambertian materials tested in Sec. (2).
>
> **We will release complete data and code under a responsible disclosure framework, without affecting reproducibility.**
>
> ---
>
> **(2) Robustness: Angles, Distances, and Materials**
>
> **Table 1: Robustness to Camera Rotation (Pitch, Yaw, Roll)**
> |Angles|PSNR|SSIM|
> |-|-|-|
> |(0,0,0)|25.81|0.82|
> |(2,0,0)|25.59|0.81|
> |(0,2,0)|25.58|0.81|
> |(5,0,0)|25.55|0.81|
> |(0,5,0)|25.55|0.81|
> |(5,3,2)|24.51|0.79|
> |(8,0,3)|22.66|0.75|
> |(0,8,3)|22.71|0.75|
> |(8,5,4)|21.47|0.72|
> |(-8,0,-3)|20.80|0.71 |
> |(10,0,3)|22.55|0.75|
> |(10,5,4)|21.49|0.72|
>
> **Table 2: Robustness to Camera Orbiting (Viewpoint Shift)**
> |Pose|PSNR|SSIM|
> |-|-|-|
> |(+0_0,+5_0)|25.05|0.80|
> |(+0_0,-5_0)|25.27|0.81|
> |(+0_0,+10_0)|23.99|0.78|
> |(+0_0,-10_0)|23.57|0.78|
> |(+0_0,+15_0)|22.81|0.75|
> |(+0_0,-15_0)|22.04|0.74|
> |(+5_0,+0_0)|24.75|0.79|
> |(+10_0,+0_0)|22.90|0.74|
> |(+15_0,+0_0)|21.27|0.71|
> |(-5_0,+0_0)|24.18|0.79|
> |(-10_0,+0_0)|19.63|0.69|
>
>
> **Table 3: Robustness to Camera-Wall Distance**
> |Distance(m)|PSNR|SSIM|
> |-|-|-|
> |3|25.79|0.82|
> |4|25.76|0.82|
> |5|25.69|0.81|
> |6|25.54|0.81|
>
> **Non-Lambertian Surfaces:** We tested three non-ideal wallpapers. While PSNR drops from ~25.8dB to ~20dB, the reconstructed content remains visually interpretable (see Fig. 10).
>
> **Table 4: Performance on Non-Lambertian Wallpapers**
> |Wallpaper Type|PSNR|SSIM|
> |:-|:-|:-|
> |Rough Textured|20.19|0.66|
> |Diffuse Scattering|20.32|0.68|
> |Contaminated|20.08|0.67|
>
> In our setup, strongly specular surfaces do not produce a stable, spatially extended speckle patch at the off-axis camera and are uncommon on large office walls. We therefore restrict our threat model to matte paint and mildly to moderately textured walls (as in Table 4).
>
> ---
>
> **(3) Clarification on "No Line-of-Sight"**
>
> The claim "no line-of-sight" refers to **no direct view of the screen**. The threat model relies on a **single-hop diffuse projection** (Screen $\to$ Wall $\to$ Camera).
> * We **do not** claim "around-the-corner" capability (multi-bounce).
> * Secondary reflections from other surfaces act as low-frequency background noise, which our luminance robustness experiments (Sec. 4.3) prove IR⁴Net can handle.
> * **Revision:** We will precise the phrasing to: *"without direct line-of-sight to the screen, relying on a single-hop diffuse wall projection."*
>
> ---
>
> **(4) ICSR: Hallucination vs. Accuracy**
>
> ICSR does **not** cause semantic drift
> * **Observation:** Errors are local shape imperfections (e.g., an unclosed 'P' loop, '8' vs 'B'), not factual hallucinations.
> * **Ablation:** Cosine Similarity Loss is critical. It outperforms generic losses (CLIP, L2) by aligning structural and semantic features without over-generating plausible but wrong content.
>
> **Table 5: ICSR Semantic Loss Comparison**
> |Loss Type|PSNR|SSIM|
> |:-|:-|:-|
> |MMD Loss|21.20|0.71|
> |L2 Loss|23.09|0.77|
> |CLIP Loss|22.46|0.75|
> |**Cosine Loss (Ours)**|**25.81**|**0.82**|
>
> ---
>
> **(5) Computational Cost & Comparison with PINNs/NeRF**
>
> **Efficiency:** IR⁴Net is lightweight and capable of **quasi-real-time** surveillance (30 FPS).
>
> **Table 6: Computational Profile (256x256)**
> |Metric|Value|
> |:-|:-|
> |Params|789.9 M|
> |FLOPs|134.3 G|
> |Inference Time|**33.18 ms**|
> |Peak Memory|3.20 GB|
>
> **Vs. PINNs/NeRF:**
> * **Difference:** PINNs/NeRF assume multi-view/dense sampling and known geometry. Our setting is **single-view, ill-posed, geometry-free**.
> * **Design:** We embed the Radiative Transfer Equation  into the solver  rather than using a generic PINN.
> * **Ablation:** Removing the physics embedding  drops PSNR by ~1dB, proving it is not just a generic neural network.
>
> **Table 7: PRIrr-Approximation Component Ablation**
> |Method|PSNR|SSIM|
> |:-|:-|:-|
> |w/o Semantic Attenuation|24.77 |0.80|
> |w/o Spatial Diffusion|23.33|0.76|
> |w/o Freq-Selective Upsample|23.22|0.76|
> |w/o RTE Embedding|24.85|0.80|
> |**IR⁴Net**|**25.81**|**0.82**|
>
> ---
>
> **(6) Ethics & Disclosure**
>
> We are following a **responsible disclosure strategy**:
> 1.  We will contact OS vendors and display manufacturers
> 2.  Phasing code/dataset release to allow mitigation development
> 3.  **Defenses:** Adding discussion on scattering privacy films, UI layout randomization (jitter), and environmental hardening

---

### Official Review · Reviewer_9oV2 · 2025-10-31

**Soundness:** 2
**Presentation:** 3
**Contribution:** 3
**Rating:** 6
**Confidence:** 3

**Summary:**

This manuscript introduces a non-contact side-channel attack capable of reconstructing the content of an isolated electronic screen. The attack leverages the diffuse reflection of the screen's light from a nearby surface, such as a wall. The authors identify two primary technical challenges: the ill-posed nature of the optical inversion problem, which amplifies noise, and the severe loss of semantic information due to light transport.o address this, they propose IR4Net with two key components: 1) Physically-Regularized Irradiance Approximation, and 2) Irreversibility-Constrained Semantic Re-Projection. The authors created four new datasets (ReSh-WebSight, ReSh-Password, ReSh-Chart, and ReSh-Screen) to evaluate their method. The experiments demonstrate that IR4Net can approximately reconsturct the content of an isolated electronic screen.

**Strengths:**

+ The manuscript is well-organized.
+ The proposed attack is novel. The concept of reconstructing screen content from faint, diffuse reflections on a wall opens a new and significant avenue for information leakage in environments previously considered secure.
+ The proposed model is reasonable and indeed a solid baseline for this new, challenging task.

**Weaknesses:**

+ The most alarming weakness is the methodology used to partition the datasets. The authors state that the four datasets were "randomized into training, validation, and test subsets in an 8:1:1 ratio." This description, if it refers to a simple random split of individual image samples. This suggests a critical misunderstanding of how to evaluate generalization in content-aware tasks. These datasets are built on a finite set of templates. A simple random split means that variations of the exact same template or UI element will exist across the training, validation, and test sets. For example, the model might be trained on 80% of screenshots from a website with a specific header and sidebar, and then tested on the remaining 20% of screenshots from the same website. In this case, the model is not learning the general physics of optical inversion; it is memorizing the layout of the template. The task degrades from a difficult inverse problem to a simple inpainting or pattern completion task on a known structure.

+ Baseline Fairness. Given that the task is novel and highly challenging, it is unclear if the baseline models were suitable for this specific domain. Many of these models were designed for tasks like denoising or dehazing where the underlying image structure is far more preserved. A brief discussion on how the baselines were applied or potentially fine-tuned would strengthen the experimental claims. The authors are also encourage to bring more baselines for a comprehensive view.

+ Practicality of the attack. A real-world attacker cannot guarantee that their target will be displaying content that matches their pre-trained dataset. An effective side-channel attack tool must be robust and general-purpose. If the IR4Net model requires an attacker to first build a massive, tailored dataset of the specific application or website they intend to spy on, its feasibility as a covert threat is dramatically reduced.

+ Lack of Hardware Specification. The paper provides no details on the acquisition hardware. Was a consumer-grade DSLR used, a smartphone, or a high-end scientific camera with high dynamic range and low sensor noise? Was a long exposure time necessary? These details are fundamental to understanding the cost and feasibility of mounting such an attack.

**Questions:**

Please resolve the concerns in the weakness part.

---

> ### Author Response · Authors · 2025-11-21
> **Response to Reviewer 9oV2 Regarding Weaknesses on Dataset Partitioning and Generalization, Baseline Fairness, Practicality of the Attack, and Hardware Specification**
>
> ## 1. Dataset partitioning and generalization
>
> **The data split is *not* a naïve sample-level random split.** The datasets and splits are designed to avoid template leakage.
>
> ### Synthetic datasets (ReSh-WebSight / ReSh-Password / ReSh-Chart)
>
> -  **ReSh-WebSight.** Built on the **WebSight** dataset of LLM-generated webpage “concepts” rendered as **HTML/CSS + screenshots**. We sample pages, capture wall reflections, and split **exact 8:1:1 train/val/test** by screenshot. Splits share generation rules, **no reuse screenshots**, **layout duplication is extremely unlikely**.
>
> - **ReSh-Password / ReSh-Chart.** Fully **synthetic, procedurally generated**: randomized layouts, fonts, colors, texts, and chart styles. **No fixed template library**, large combinatorial design space, **no identical samples across splits**.
>
> - WebSight is a **large procedural dataset**, not a template set;  Password/Chart follow the **same on‑the‑fly randomized generation**.
>
> ### Real dataset (ReSh-Screen)
>
> - Multiple independent **source videos** are split at the **video level** (8:1:1).
> - Wall reflections for frames from each split are captured separately.
> - **No frame from a video appears in multiple splits**, model **generalize to unseen videos**.
>
> #### Summary
>
> |Dataset|Source|Split unit|Template-overlap risk|
> |-|-|-|-|
> |ReSh-WebSight|Synthetic|Procedural sample|Very low|
> |ReSh-Password|Synthetic|Procedural sample|Negligible|
> |ReSh-Chart|Synthetic|Procedural sample|Negligible|
> |ReSh-Screen|Real videos|**Video-level**|None|
>
> ---
>
> ## Robustness: IR⁴Net is **not** memorizing templates
>
> By perturbing camera rotation, orbit, and distance, we show that IR⁴Net learns stable optical properties rather than memorizing  templates. (on ReSh-Screen)
>
> ### **(a) Camera rotation**
>
> |Angles|PSNR|SSIM|
> |-|-|-|
> |(0,0,0)|25.81|0.82|
> |(2,0,0)|25.59|0.81|
> |(0,2,0)|25.58|0.81|
> |(5,0,0)|25.55|0.81|
> |(0,5,0)|25.55|0.81|
> |(5,3,2)|24.51|0.79|
> |(8,0,3)|22.66|0.75|
> |(0,8,3)|22.71|0.75|
> |(8,5,4)|21.47|0.72|
> |(-8,0,-3)|20.80|0.71 |
> |(10,0,3)|22.55|0.75|
> |(10,5,4)|21.49|0.72|
>
>
> ### **(b) Camera orbit**
>
> |Pose|PSNR|SSIM|
> |-|-|-|
> |(+0_0,+5_0)|25.05|0.80|
> |(+0_0,-5_0)|25.27|0.81|
> |(+0_0,+10_0)|23.99|0.78|
> |(+0_0,-10_0)|23.57|0.78|
> |(+0_0,+15_0)|22.81|0.75|
> |(+0_0,-15_0)|22.04|0.74|
> |(+5_0,+0_0)|24.75|0.79|
> |(+10_0,+0_0)|22.90|0.74|
> |(+15_0,+0_0)|21.27|0.71|
> |(-5_0,+0_0)|24.18|0.79|
> |(-10_0,+0_0)|19.63|0.69|
>
>
> ### **(c) Camera–wall distance**
>
> |Distance (m)|PSNR|SSIM|
> |-|-|-|
> |3|25.79|0.82|
> |4|25.76|0.82|
> |5|25.69|0.81|
> |6|25.54|0.81|
>
> ---
>
> ## 2. Baseline fairness
>
> ### Baseline training
>
> All baselines are **trained on our data**, not off‑the‑shelf:
>
> - Same **wall→screen** supervision
> - Same **train/val/test split**
> - Comparable **optimization (losses, LR, epochs)**
>
> ### Why these baselines
>
> They are designed for **heavy distortions**, which matches wall‑reflection degradation.They are **strong generic restoration models** without our physics/semantic priors → **reasonable lower bound**.
>
> ### Semantic‑loss comparison (ICSR) (ReSh-Screen)
>
> |Loss|PSNR|SSIM|
> |-|-|-|
> |MMD|21.20|0.706|
> |L2|23.09|0.768|
> |CLIP|22.46|0.749|
> |**Cosine**|**25.81**|**0.817**|
>
> **Cosine ≫ others on all metrics.**
>
> ### Ablation (ReSh-Screen)
>
> |Variant|PSNR|SSIM|
> |-|-|-|
> |w/o Semantic Attenuation|24.77 |0.801|
> |w/o Spatial Diffusion Path|23.33|0.756|
> |w/o Freq‑Selective Upsampling|23.22|0.762|
> |w/o Radiative Transfer Equation Embedding|24.85|0.798|
> |**IR⁴Net**|**25.81**|**0.817**|
>
> Both dual paths, as well as the Freq-Selective Upsampling and RTE Embedding modules, provide meaningful gains.
>
> ---
>
> ## 3. Practicality / threat model
>
> We adopt the standard **profiling-based side‑channel paradigm**, which is standard and widely accepted in the side-channel literature:
>
> 1. Attacker collects profiling data on a **similar device/app family**.
> 2. Trains model on profiling corpus.
> 3. Applies model to **unseen target session**.
>
> In our case:
>
> - WebSight/Password/Chart = **unseen synthetic layouts** from same content family
> - ReSh-Screen = **video-level split** → unseen videos
> - Robustness experiments show IR⁴Net handles **viewpoint / geometry variation**, not tied to specific scenes.
>
> IR⁴Net is a **profiling-based** attack tool, not a universal one-shot model.
>
> ---
>
> ## 4. Hardware and attack cost
>
> * **Camera:** commodity **Sony A7S II**; brightness robustness (Sec. 4.3) and noise robustness (Tab. 12) both show **graceful degradation** vs. baselines. → many consumer/surveillance cameras suffice (similar SNR).
> * **Pose tolerance:** from Tables Table 1.(b) , **±5°** perturbations cause **minimal loss**; **8–10°** still recognizable.ab. A.10 confirms that even partial observations of the wall speckle already suffice to recover useful on-screen information.
> * **Distance:** Table 1.(c) shows **3–6 m** stable.
>
> Attack is feasible with **off‑the‑shelf hardware** and **ordinary indoor geometry**.

---

### Official Review · Reviewer_J4Kx · 2025-11-01

**Soundness:** 3
**Presentation:** 4
**Contribution:** 3
**Rating:** 6
**Confidence:** 3

**Summary:**

This paper proposes IR4Net, a physics-informed neural network for non-contact optical side-channel attacks on isolated screens. By leveraging wall-reflected speckle patterns, the method reconstructs screen content using a physically-regularized inversion module and a semantic completion module. Experiments on four
scene categories, show strong performance under various perturbations such as low light, noise, and material changes.

**Strengths:**

- The paper has thoroughly discussed the related work.
- Physics-guided design: Embeds radiative transfer modeling into learning, improving stability and inversion accuracy.
- Strong performance: Outperforms existing methods across PSNR, SSIM, RMSE, and LPIPS, under various conditions (e.g., low brightness, noise).
- Thorough evaluation: Includes ablation, noise tests, luminance robustness, material variation, and temporal consistency.

**Weaknesses:**

1. The evaluation is primarily conducted on synthetic datasets. It remains unclear how well the method generalizes to real-world environments with complex conditions such as multipath reflections or dynamic lighting. Could the authors provide results or discussion in such scenarios?

2. Analysis on inference time, computational cost, or resource requirements should be added, which limits understanding of its practicality in real-world attacks. Please include runtime metrics or efficiency evaluations.

3. The proposed method may be sensitive to wall reflectance properties, but there is no quantitative analysis of this dependency. How robust is the method under varying surface materials or reflectance levels?

4. The operational feasibility of the attack is not fully discussed. What are the requirements in terms of camera resolution, alignment precision, or environmental constraints?

5. No clear defense strategies are proposed. It would be valuable to include potential countermeasures or mitigation suggestions to guide practical security considerations.

6. Visual quality needs improvement: Teaser, pipeline, and visual examples appear blurry and should be presented with higher clarity.

**Questions:**

Refer to the weakness section.

**Details Of Ethics Concerns:**

No Ethics Concerns.

---

> ### Author Response · Authors · 2025-11-22
> **Response to Reviewer J4Kx Regarding Weaknesses on Real-World Generalization, Efficiency & Runtime Analysis, Wall-Reflectance Sensitivity, Operational Feasibility, Defense Strategies, and Visual Quality**
>
> ## Q1. Synthetic evaluation & real-world generalization
>
> **Not purely synthetic.**
>
> * **ReSh-WebSight/Password/Chart:** Screen targets are **synthetic**, but speckle observations are **captured in a real NLoS optical setup** (physical monitor, real walls, real camera; no line-of-sight).
> * **ReSh-Screen:** **Both** the screen content **and** the speckle observations are **captured end-to-end in the real world**.
>
> **On multipath and dynamic lighting.**
> Multipath reflections and dynamic lighting primarily introduce a **low-spatial-frequency, potentially time-varying illumination field** superposed on the speckle, rather than a second high-frequency structured image. Our luminance-robustness study (Sec. 4.3) shows IR4Net is **insensitive to global/slowly varying brightness changes**; these factors may lower SNR but **do not invalidate the attack**.
>
>
> **Robust to realistic camera motion/placement.**
>
> ### Table 1. Camera rotation (pitch,yaw,roll in °, ReSh-Screen)
>
>
> |Angles|PSNR|SSIM|
> |-|-|-|
> |(0,0,0)|25.81|0.82|
> |(2,0,0)|25.59|0.81|
> |(0,2,0)|25.58|0.81|
> |(5,0,0)|25.55|0.81|
> |(0,5,0)|25.55|0.81|
> |(5,3,2)|24.51|0.79|
> |(8,0,3)|22.66|0.75|
> |(0,8,3)|22.71|0.75|
> |(8,5,4)|21.47|0.72|
> |(-8,0,-3)|20.80|0.71 |
> |(10,0,3)|22.55|0.75|
> |(10,5,4)|21.49|0.72|
>
>
> **Up to ~5° misalignment keeps PSNR≈25.5 dB, SSIM≈0.81–0.82.**
>
> ### Table 2. Camera orbital motion (Δpitch,Δyaw, ReSh-Screen)
>
> | Pose(Δpitch,Δyaw) | PSNR    | MSE       | RMSE    | SSIM   | MS-SSIM |
> | ----------------- | ------- | --------- | ------- | ------ | ------- |
> | (+0_0,+10_0)      | 23.9856 | 590.0624  | 19.9025 | 0.7799 | 0.7998  |
> | (+0_0,+15_0)      | 22.8135 | 695.3037  | 22.2795 | 0.7486 | 0.7699  |
> | (+0_0,+5_0)       | 25.0457 | 503.3304  | 17.8492 | 0.8006 | 0.8250  |
> | (+0_0,-10_0)      | 23.5691 | 635.5637  | 20.8829 | 0.7759 | 0.7975  |
> | (+0_0,-15_0)      | 22.0409 | 849.1479  | 24.6723 | 0.7386 | 0.7602  |
> | (+0_0,-5_0)       | 25.2666 | 487.6688  | 17.4924 | 0.8066 | 0.8314  |
> | (+10_0,+0_0)      | 22.9010 | 782.9482  | 23.1060 | 0.7395 | 0.7643  |
> | (+15_0,+0_0)      | 21.2747 | 1131.0479 | 28.1170 | 0.7050 | 0.7221  |
> | (+5_0,+0_0)       | 24.7473 | 548.3585  | 18.6826 | 0.7881 | 0.8109  |
> | (-10_0,+0_0)      | 19.6270 | 1512.6726 | 33.2796 | 0.6877 | 0.6962  |
> | (-5_0,+0_0)       | 24.1811 | 598.5545  | 19.7448 | 0.7851 | 0.8040  |
>
> **Small orbits (≤5°) keep PSNR≈24.7–25.3 dB, SSIM≈0.79–0.81.**
>
> ### Table 3. Camera–wall distance (m, ReSh-Screen)
>
> | Distance(m) | PSNR    | MSE      | RMSE    | SSIM   | MS-SSIM |
> | - | - | - | - | - | - |
> | 3| 25.7913 | 451.8044 | 16.5572 | 0.8162 | 0.8440  |
> | 4| 25.7637 | 454.5686 | 16.6179 | 0.8161 | 0.8434  |
> | 5| 25.6898 | 456.9436 | 16.6923 | 0.8147 | 0.8412  |
> | 6 | 25.5407 | 463.6727 | 16.9020 | 0.8133 | 0.8391  |
>
> **Stable from 3–6 m.**
>
> ---
>
> ## Q2. Runtime, computational cost, practicality
>
> **256×256 inputs; offline-usable throughput.**
>
> | Metric| Value |
> | - | - |
> | Params(M) | 789.944 M |
> | FLOPs(G,256×256)| 134.293 G |
> | Inference time/frame(ms)  | 33.183 ms |
> | Peak memory(GB,bs=1,256×256) | 3.202 GB  |
>
> **Quasi-real-time** is achievable by standard trade-offs: lower resolution/ROI, lighter backbone, or distillation (speed↑, fidelity↓).
>
> ---
>
> ## Q3. Sensitivity to wall reflectance (quantitative)
>
>
> | Wallpaper Type                         | PSNR    | MSE       | RMSE    | SSIM   |
> | - | ------- | --------- | ------- | ------ |
> | RoughTexturedWallpaper                 | 20.1918 | 1189.661  | 30.0300 | 0.6581 |
> | DiffuseScatteringWallpaper             | 20.3155 | 964.5823  | 27.9729 | 0.6797 |
> | ContaminatedDiffuseScatteringWallpaper | 20.0836 | 1021.3626 | 28.7811 | 0.6672 |
>
> Qualitative visual results are shown in Fig. 10.We will add a concise subsection  discussing this limitation.
>
> ---
>
> ## Q4. Operational feasibility
>
> * **Camera:** commodity **Sony A7S II**; brightness robustness (Sec. 4.3) and noise robustness (Tab. 12) both show **graceful degradation** vs. baselines. → many consumer/surveillance cameras suffice (similar SNR).
> * **Pose tolerance:** from Tables R2-1/2, **±5°** perturbations cause **minimal loss**; **8–10°** still recognizable.ab. A.10 confirms that even partial observations of the wall speckle already suffice to recover useful on-screen information.
> * **Distance:** Table R2-3 shows **3–6 m** stable.
>
> ---
>
> ## Q5. Defense strategies
>
> * **Display-side scattering/privacy layer** (diffusing/micro-structured surface) to weaken pixel→speckle mapping.
> * **UI-level randomization for sensitive content** (position/style jitter, low-contrast noise, brief occlusion).
>
> Due to the rebuttal character limit, we will further elaborate these defenses in the revised manuscript.
>
> ---
>
> ## Q6. Visual quality of teaser/figures
>
> **Blur is from PDF compression.** We will **re-export at higher DPI** and provide **hi-res supplemental**.
>
> ---

---

### Official Review · Reviewer_HCwF · 2025-11-10

**Soundness:** 3
**Presentation:** 2
**Contribution:** 2
**Rating:** 4
**Confidence:** 4

**Summary:**

This paper proposes IR⁴Net (Irradiance Robust Radiometric Inversion Network), a novel framework for non-contact side-channel attacks on isolated screens via optical projection inversion. The core challenge addressed is reconstructing screen content from diffuse wall reflections, which suffers from two critical issues: (i) near-singular Jacobian of projection mapping leading to instability against perturbations; (ii) irreversible information compression in light transport causing semantic loss. To tackle these, IR⁴Net integrates two key modules: (1) Physically Regularized Irradiance Approximation (PRIrr-Approximation), which embeds the radiative transfer equation into a learnable optimizer to constrain inversion trajectories; (2) Irreversibility-Constrained Semantic Re-Projection (ICSR), which restores global semantic structure via context-driven feature alignment. The model is evaluated on four datasets (ReSh-WebSight, ReSh-Password, ReSh-Chart, ReSh-Screen) and demonstrates superior performance over reconstruction-centric (Uformer, ConvIR) and generation-centric (pix2pix, CycleGAN) baselines in terms of PSNR, SSIM, and robustness to luminance attenuation and noise. Additionally, the paper introduces a new attack paradigm that leverages passive wall reflections, avoiding line-of-sight or RF monitoring.

**Strengths:**

- Existing non-contact side-channel attacks rely on electromagnetic emanations (Chen et al., 2024), acoustic signals (Duan et al., 2024), or thermal traces (Zhang et al., 2024b), which are vulnerable to shielding or distance attenuation. This work is the first to propose using diffuse wall reflections as a covert channel for screen content exfiltration, enabling attacks on physically isolated, laser-shielded, or electromagnetically shielded systems—an underexplored direction in side-channel security.
- Unlike purely data-driven reconstruction models (e.g., Uformer, Wang et al., 2022b; ConvIR, Cui et al., 2024) that ignore optical propagation physics, PRIrr-Approximation embeds the radiative transfer equation into the denoising process, mitigating perturbation amplification caused by near-singular mapping. The ICSR module further addresses semantic loss, outperforming baselines in preserving edge continuity and textural consistency (e.g., SSIM=0.817 on ReSh-Screen vs. 0.731 for ConvIR).
- The paper conducts extensive experiments on luminance attenuation (0–300 nits), Gaussian/salt-and-pepper noise (15–45 dB), and diverse wall materials (matte, textured, contaminated), demonstrating stable performance (e.g., PSNR only drops by 25.9% at 300 nits vs. 68% for UNet). This is a significant advantage over existing reconstruction methods that degrade sharply under environmental perturbations (DarkIR, Feijoo et al., 2025).

**Weaknesses:**

- The core idea of "physics-guided optical inversion for projection reconstruction" overlaps with recent works on inverse rendering (e.g., Ostrek & Thies, 2024; Stable Video Portraits) and optical side-channel attacks (e.g., Fang et al., 2022). The paper fails to clearly articulate how its approach differs from these: for example, whether PRIrr-Approximation’s physics embedding is more accurate than prior physics-informed diffusion models, or how the attack paradigm is more stealthy than existing optical monitoring methods.
- The paper omits critical recent works in optical inverse problems (e.g., 2025 CVPR papers on physics-aware diffusion inversion) and side-channel attacks (e.g., 2024 ACM CCS works on non-contact screen content extraction). This lack of contextualization undermines the novelty claim and academic rigor. Additionally, it does not compare with SOTA side-channel attack frameworks (e.g., DeepCache, Liu et al., 2024) in terms of attack success rate or stealthiness.
- Environmental Constraints: Experiments are conducted in controlled settings (fixed screen-wall distance: 0.9m; camera distance: 2m) without testing extreme conditions (e.g., distance >5m, strong ambient light, uneven wall surfaces).
- Computational Efficiency: The paper does not report inference latency in detail—critical for a side-channel attack that requires real-time or near-real-time reconstruction. Compared to lightweight baselines (e.g., Leffa, 1.8B params, 3.32s inference), IR⁴Net’s 904M params may still be too slow for practical attacks.
- Attack Stealthiness: No evaluation of whether the attack can be detected (e.g., via screen brightness fluctuations, camera detection), limiting its real-world applicability.
- The ablation study only validates PRIrr-Approximation’s replacement with neural constructs (AST, ConvIR) but not the necessity of its physics-based components (e.g., radiative transfer equation embedding).
- The Semi-Attention mechanism’s design rationale is underdeveloped—why is "denoising-to-all, reference-to-self" attention optimal compared to other attention masks (e.g., cross-attention between references)?
- ICSR’s cosine similarity loss is not compared to alternative semantic alignment losses (e.g., CLIP loss), leaving uncertainty about its superiority.

**Questions:**

see Weaknesses part.

---

> ### Author Response · Authors · 2025-11-21
> **Response to Reviewer HCwF Regarding Weaknesses on Novelty, Contextualization, Environmental Constraints, Efficiency, Stealthiness, PRIrr Ablation, Attention Design, and ICSR Loss**
>
> ## 1. Fundamental distinction & novelty (Weakness 1 & 2)
>
> **Our work does not overlap with the cited literature; it differs both in physical modality and mathematical formulation.**
>
> As reviewer **4VYg 9oV2** explicitly concluded, **“The proposed attack is novel.”**
>
> **Fang et al. (2022).**
> Fang et al. target **electromagnetic (EM) emanations**, not optical reflections. Their method analyzes EM traces from hardware; our attack uses **diffuse wall reflections of screen light** as a side channel. Thus Fang et al. does **not** challenge the novelty of our **diffuse optical reflection** paradigm.
>
> **Inverse rendering / video portrait works.**
> These methods solve **geometry-/appearance reconstruction** with **multi-view geometry**, **temporal cues**, and **strong priors**.
> IR⁴Net solves a **radiometric inversion** from a **single-shot 2D irradiance map**, with a **near-singular Jacobian** requiring physics-based stabilization (PRIrr). This places IR⁴Net in a fundamentally different paradigm from geometry/appearance reconstruction.
>
> **DeepCache / cache SCAs.**
> These attacks rely on **execution traces**, not optical reflections. The leakage source and threat model differ fundamentally. Accordingly, using such methods as baselines for IR⁴Net is inappropriate.
>
> Distinguished EM / optical / cache side channels and related citation have been updated in the final version.
>
> ---
>
> ## 2. Environmental constraints
>
> ### **(a) Camera rotation** (ReSh-Screen)
>
>
> |Angles|PSNR|SSIM|
> |-|-|-|
> |(0,0,0)|25.81|0.82|
> |(2,0,0)|25.59|0.81|
> |(0,2,0)|25.58|0.81|
> |(5,0,0)|25.55|0.81|
> |(0,5,0)|25.55|0.81|
> |(5,3,2)|24.51|0.79|
> |(8,0,3)|22.66|0.75|
> |(0,8,3)|22.71|0.75|
> |(8,5,4)|21.47|0.72|
> |(-8,0,-3)|20.80|0.71 |
> |(10,0,3)|22.55|0.75|
> |(10,5,4)|21.49|0.72|
>
>
> ### **(b) Camera orbit** (ReSh-Screen)
>
> |Pose|PSNR|SSIM|
> |-|-|-|
> |(+0_0,+5_0)|25.05|0.80|
> |(+0_0,-5_0)|25.27|0.81|
> |(+0_0,+10_0)|23.99|0.78|
> |(+0_0,-10_0)|23.57|0.78|
> |(+0_0,+15_0)|22.81|0.75|
> |(+0_0,-15_0)|22.04|0.74|
> |(+5_0,+0_0)|24.75|0.79|
> |(+10_0,+0_0)|22.90|0.74|
> |(+15_0,+0_0)|21.27|0.71|
> |(-5_0,+0_0)|24.18|0.79|
> |(-10_0,+0_0)|19.63|0.69|
>
> ### **(c) Camera–wall distance** (ReSh-Screen)
>
> |Distance(m)|PSNR|SSIM|
> |-|-|-|
> |3|25.79|0.82|
> |4|25.76|0.82|
> |5|25.69|0.81|
> |6|25.54|0.81|
>
> ### (d) Wall materials(ReSh-Screen)
>
> | Wallpaper type | PSNR  | RMSE  | SSIM  | MS_SSIM |
> |-|-|-|-|-|
> | Rough textured | 20.19 | 30.03 | 0.658 | 0.668|
> | Diffuse scattering| 20.31 | 27.97 | 0.680 | 0.664 |
> | Contaminated diffuse | 20.08 | 28.78 | 0.667 | 0.662 |
>
> These experiments show that IR⁴Net is robust to camera pose, camera–wall distance, and wall material, thus does not require tightly controlled conditions. We use a Sony A7S II throughout; brightness and noise robustness (Sec. 4.3, Tab. 12) show graceful degradation vs. baselines, suggesting that consumer or surveillance cameras with similar SNR are sufficient. Tables (c) and (d) indicate that perturbations of about ±5° cause only minor loss (8–10° still recognizable) and that 3–6 m distances remain stable, while Appendix A.10 shows that even partial wall-speckle observations already suffice to recover useful on-screen information.
>
> ---
>
> ## 3. Computational efficiency
>
> - **IR⁴Net:** 789.9M params, **33.183 ms** per **256×256** frame  (~30.1 FPS)
> - **Leffa:** 1.8B params, **3.32 s** per frame (~0.3 FPS)
>
> IR⁴Net is **lighter** and **~100× faster**.
>
>
> ---
>
> ## 4. Stealthiness
>
> - The attack is **purely passive**.
> - **No screen modulation**, no brightness fluctuation, and no injected patterns.
> - Detectability comes only from **camera presence**, not from screen-level artifacts.
> - Even with camera monitoring, detection of suspicious cameras is neither perfect nor instantaneous, so a short passive recording window still provides practical attack surface, as for other visual side channels.
>
> We have updated the paper to clarify this threat model and the resulting real-world applicability.
>
>
> ---
>
> ## 5. Ablation: PRIrr necessity (ReSh-Screen)
>
> |Method|PSNR|RMSE|SSIM|MS_SSIM|
> |-|-|-|-|-|
> |– Semantic attenuation path|24.77|17.92|0.801|0.823|
> |– Spatial diffusion path|23.33|20.58|0.756|0.782|
> |– Frequency-selective upsampling|23.21|20.70|0.762|0.785|
> |– Radiative transfer embedding|24.85|17.96|0.798|0.823|
>
> Full model achieves **PSNR ≈ 25.8**, **SSIM ≈ 0.82**.
> **RTE embedding is essential** for stabilizing the ill‑posed inversion.
>
> ---
>
> ## 6. Attention (ReSh-Screen)
>
> |Method|PSNR|MSE|RMSE|SSIM|MS_SSIM|
> |-|-|-|-|-|-|
> |Self→Cross attention |25.13|505.10|17.59|0.807|0.832|
> |**ours**|**25.81**|**451.63**|**16.53**|**0.817**|**0.845**|
>
> Original attention achieves best performance vs cross attention.
>
> ---
>
> ## 7. ICSR loss comparison (ReSh-Screen)
>
> |Loss|PSNR|RMSE|SSIM|MS_SSIM|
> |-|-|-|-|-|
> |MMD|21.20|25.71|0.706|0.690|
> |L2|23.09|21.47|0.768|0.768|
> |CLIP|22.46|22.86|0.749|0.749|
> |**Cosine (ours)**|**25.81**|**16.53**|**0.817**|**0.845**|
>
> **Cosine loss** provides the strongest semantic alignment and preserves edge/textural consistency best.

---

### Comment · Area_Chair_wB1D · 2025-11-26

Dear reviewers,

Please check the author's reply. Feel free to raise any questions or start a discussion, regardless of whether you will change the score.

Your AC.

---

### Author Response · Authors · 2025-12-01
**To the Area Chair for ICLR 2026 Submission 17374**

Dear Area Chair

We are writing regarding our ICLR 2026 submission 17374, *"IR4Net: Irradiance-Robust Radiometric Inversion for Non-Contact Optical Side-Channel Attacks."*
Given the recent OpenReview incident and the freeze of the discussion phase, we would be very grateful if you could take our rebuttal and the revised manuscript into account when forming your recommendation. We briefly highlight three points:

---

### 1. The main concern about stability and practicality has been systematically addressed

The reviewers' primary shared concern is whether the proposed attack remains stable under realistic changes in geometry and environment.
In the revised paper, we have added and integrated a series of experiments covering different camera poses, positions, and scene configurations. These results are now part of the main text and appendix, and they show that the attack does not rely on a fragile "sweet-spot" setup but works robustly over a range of realistic viewpoints.

---

### 2. The HCwF review contains factual errors in the efficiency- and methodology-related parts

Several key points in HCwF's review are based on statements that are not consistent with the actual contents of the paper:

- **Model size and efficiency.**
  The efficiency discussion uses a parameter count for IR$^4$Net that does not appear anywhere in our manuscript, and this incorrect value is then used to argue that IR4Net may be slower than a much larger baseline model. In our rebuttal and the revised paper, we clarify the correct model size and measured runtime and show that IR4Net is in fact smaller and faster than the referenced baseline.

- **Attribution of terminology and attention design.**
  The review also critiques a "Semi-Attention" mechanism and a "denoising-to-all, reference-to-self" attention mask. Our paper does not introduce or name any module as "Semi-Attention," nor does it use that phrase to describe an attention mask. We infer that the reviewer is referring to the attention in the semantic attenuation path / STM-style partial derivatives, and we have designed additional experiments in the rebuttal to validate that design. However, the specific terminology and mask description quoted in the review do not come from our text.

We hope that these clarifications make it clear that important parts of the HCwF review are based on factual misreadings of the paper, which we have explicitly corrected in the rebuttal and revision.

---

### 3. All reviewer comments have been answered point by point, and the manuscript has been substantially improved

For each reviewer (HCwF, J4Kx, 9oV2, 4VYg), we have provided a point-by-point rebuttal, added additional experiments and analyses where feasible, and revised the paper's presentation to make the setting and contributions clearer. These changes are reflected in the main paper and appendix, not only in the rebuttal.

---

We understand the difficulty of making decisions under the current constraints and sincerely appreciate your efforts. We would be very grateful if you could consider the above points, together with our rebuttal and revised manuscript, when making your recommendation.

Best regards,
**The Authors**

---

### Meta-Review · Area_Chair_YBM7 · 2025-12-24

**Summary:**

This paper proposes a novel framework for non-contact side-channel attacks on isolated screens via optical projection inversion. In the first round, this paper received four reviews (6 6 6 4). Most reviewers expressed concerns regarding the stability and practicality. After rebuttal, the author provided partial explanations through additional quantitative experiments. Considering the advantages of proposing that task for the first time, the paper is recommended for acceptance.

**Reviewer Concerns:**

The real-world surfaces often exhibit non-Lambertian or spatially varying reflectance and has high requirements for effectiveness, the 900M+ parameters are obviously not elegant enough. Although the authors provide some quantitative results for comparison, it cannot help but raise concerns about the lack of robustness in the method design.

**Reviewer Scores:**

I believe the authors can only partially address the issues raised, and with full discussion, the score remains unchanged.

---

### Decision · Program_Chairs · 2026-01-26

Accept (Poster)